# How Can Education for Sustainable Development (ESD) Be Effectively Implemented in Teaching and Learning? An Analysis of Educational Science Recommendations of Methods and Procedures to Promote ESD Goals

Werner Riess [1,*], Monika Martin [1], Christoph Mischo [2], Hans-Georg Kotthoff [3] and Eva-Maria Waltner [1]

1   ReCCE (Research Center for Climate Change Education and Education for Sustainable Development),
    University of Education Freiburg, Kunzenweg 21, 79117 Freiburg, Germany;
    monika.martin@ph-freiburg.de (M.M.); eva-maria.waltner@ph-freiburg.de (E.-M.W.)
2   Department of Psychology, University of Education Freiburg, Kunzenweg 21, 79117 Freiburg, Germany;
    mischo@ph-freiburg.de
3   Department of Pedagogy, University of Education Freiburg, Kunzenweg 21, 79117 Freiburg, Germany;
    kotthoff@ph-freiburg.de
*   Correspondence: riess@ph-freiburg.de; Tel.: +49-(0)761-682-217

**Abstract:** Education for sustainable development (ESD) has been a task assigned to schools and universities since the mid-1990s. This global movement spawned ESD research in numerous fields, including, among others, didactics and educational sciences, as well as sustainability sciences. In this article, we analyze the state of research on reliable recommendations of means (more precisely, teaching and learning methods and procedures) to promote the ESD goals. Within the framework of systematic literature analysis, we compared and evaluated 17 scientific publications from the field of ESD. Using qualitative content analysis, we scanned the 17 articles for recommendations of means of ESD and the cited evidence for their effectiveness. The findings show two groups of recommended means, differing particularly in the degree of learner autonomy and the quality of evidence for their effectiveness. We discuss possible tasks that can be derived from these findings for didactic research on ESD, and we make a suggestion for further teaching action.

**Keywords:** education for sustainable development (ESD) means; teaching and learning methods; effectiveness



## 1. Introduction

For more than two decades, ESD has been called upon to empower learners in schools and universities to promote sustainable development in a variety of contexts and settings [1,2]. Corresponding demands can be found in many education plans, curricula, and study regulations worldwide [3]. Teachers in general and secondary schools and lecturers at colleges and universities are expected to effectively implement ESD in their institutions. To cope with these tasks successfully, they especially need—in addition to extensive professional and interdisciplinary knowledge—clear answers to the following two questions: First, what characteristics does a person (learner) need to be able to shape sustainable development successfully? These characteristics should be promoted within the framework of ESD. Second, what means (i.e., learning and teaching methods and procedures that open up learning opportunities) for promoting these characteristics can be recommended and therefore used in the classroom?

In response to the second question, educators in schools and universities should especially be able to expect robust statements from research on ESD and empirical educational and teaching sciences.

A review of numerous publications for teachers on ESD (e.g., teachers' handbooks, teachers' journals, teaching concepts, recommendations for higher education didactic

courses, guidelines from ministries of education, and recommendations from NGOs) has yielded contradictory and poorly substantiated recommendations, especially with regard to the second question [2]. One might get the impression that almost all learning and teaching methods are equally effective in helping teachers successfully implement ESD, and therefore that all recommendations are equally valid. However, many of these recommendations for certain learning and teaching methods and procedures are not supported by empirical evidence but instead by merely plausible arguments or references to individual case studies. This lack of evidence is admittedly uncertain ground on which to stand when recommending means of teaching in the context of ESD. Therefore, as an interdisciplinary group of researchers (from educational psychology, empirical educational science, biology, and political didactics), we set out to collect and review studies that have specifically investigated the effectiveness of means for ESD. With our contribution, we hope to provide a first summarizing, albeit preliminary, answer to the question of effective learning and teaching methods for ESD. Furthermore, this paper aims to diagnose research gaps for further necessary research, which ultimately will contribute to a theory of effective ESD.

## 2. Theoretical and Methodological Background

Before identifying effective means, the goals and competencies of ESD must first be identified. Hence, a provisional answer must be given to the first question posed above. Identifying means and methods for achieving goals and determining whether they are effective requires first specifying the goals.

In the field of education, goals are personal characteristics to promote in learners (children, pupils, students, adults, etc.). According to Uhl [4], these personal characteristics can be subjected to (a) a normative test (are they consistent with fundamental educational ideals, e.g., maturity?) and (b) an empirical test (e.g., can these characteristics, which are initially proposed by ESD educators based on theoretical assumptions, be measured?). Reviews and critical analyses of prominent ESD goal recommendations were conducted by Rieß et al. [5] and Brundiers et al. [6]. Rieß's research group proposed sustainability competence as an overarching goal construct of ESD. Following Weinert [7], they understood sustainability competence to be the totality of cognitive abilities and skills as well as associated motivational, volitional, and social readiness needed to be able to solve sustainability-relevant problems and shape sustainable development in private, social, and institutional contexts. The concept of goal presented by this group has the following advantages over many common alternatives. The various disciplines and subjects can locate their subject-specific, sustainability-relevant goals within an overall framework for ESD, and the division into proven competency facets of educational research makes it possible to operationalize ESD goals and thus develop measurement instruments for ESD. Based on this concept, the effects of methods promoting ESD goals can now be assessed [8]. Brundiers et al. [6] took a different approach by formulating key competencies. They recommend the following seven higher-level key competencies to strive for across disciplines: Systems-thinking, Strategic-thinking, Values-thinking, Futures-thinking, Implementation, Interpersonal, and Integrated Problem-Solving. We emphasize here that some of the proposed key competencies are very general and relevant to many educational areas outside of ESD. Consequently, they can also be difficult to operationalize unless they are specified with content from the various disciplines relevant to sustainability, as suggested by the research group.

Our main research question is this: Which methods and means are recommended in the ESD literature to promote these goals, and what is the supporting evidence? We define evidence for effectiveness as sufficient empirical evidence that the use of a method or a procedure can lead to the promotion of the intended ESD goals.

Our investigation first required organizing and classifying recommendations for methods and means. A classification based on a multilevel analytical approach from educational research seemed suitable [9]. In school governance research, a rough distinction is made between a micro level (the place of interaction between teachers and learners, e.g., lessons, seminars, and training courses), a meso level (the specific educational institution,

e.g., school, and university but also extracurricular places of learning) and a macro level (e.g., ministerial requirements in the form of educational plans, curricula, examination regulations, and framework guidelines for study regulations) [10].

We decided, because of limited resources, to focus only on the means of the micro level, that is, the teaching and learning methods and procedures that are directly planned and used by the teachers to promote desired ESD goals in lessons, courses, and training programs. This approach followed the current consensus in empirical educational research that teaching and learning are crucial for learning performance [11,12]. To increase the readability of the text, we use the term *methods* for all teaching and learning methods as well as procedures. Our research questions are:

1.  What teaching and learning methods to promote ESD goals in schools and colleges are recommended in the ESD literature?
2.  What evidence is cited for the effectiveness of the recommended methods? Has the effectiveness of these methods been robustly demonstrated?

### 3. Method

We determined relevant databases to search for articles suggesting learning–teaching methods for ESD. We found ERIC (Education Resources Information Center), PSYNDEX, SCOPUS, and FIS-Bildung to be particularly suitable because they cover a wide international area (for the overall procedure, see Figure 1).

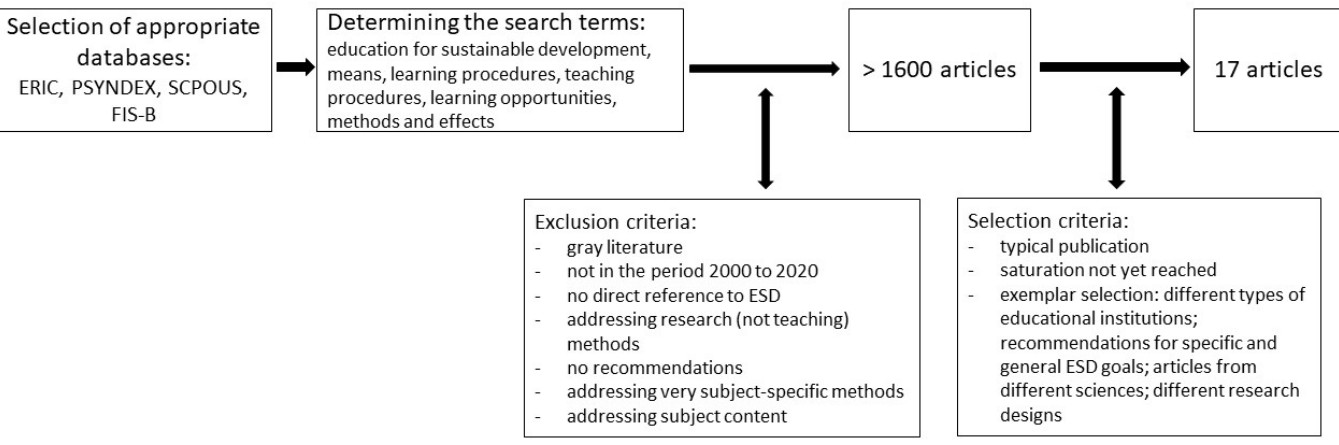

**Figure 1.** The searching process.

As search terms, we used "education for sustainable development", "means", "learning procedures", "teaching procedures", "learning opportunities", "methods", and "effects". We considered articles in scientific journals, contributions in anthologies, monographs, and doctoral theses. Gray literature was excluded. The search was limited to the period from January 2000 to December 2020. We excluded publications that (a) had no direct reference to ESD; (b) addressed research methods (not teaching methods) as methods; (c) only described means, procedures, and methods in a descriptive manner and did not make recommendations; or (d) addressed very subject-specific methods (e.g., procedures for investigating water bodies) or subject content as a means of promoting ESD goals. Given the high number of sources (N > 1600), we decided to include only one or a few typical publications in the analysis at a time and to stop after reaching conceptual saturation (i.e., an increase by one in the sample reveals no additional types of means or methods recommendations) [13]. We therefore followed the principle of exemplariness because we intended to not make representative statements.

We chose the following criteria for exemplar selection:

- Means and methods recommendations for the different types of schools as well as colleges and universities (including adult education; recommendations for the design of ESD at extracurricular learning sites were left out);
- Means and methods recommendations for specific (e.g., systems-thinking) and general ESD goals (e.g., shaping competence);
- Articles from different educational and sustainability sciences;
- Different research designs (e.g., quasi-experimental, experimental, case studies).

In the end, 16 articles in professional journals or anthologies and one monograph formed the source corpus. They were written by authors who were only or predominantly rooted in subject didactics (e.g., biology, geography; 5 articles), educational sciences (6 articles) and sustainability sciences (6 articles). The evaluation of the articles was based on qualitative content analysis according to Mayring [14]. Qualitative content analysis is a method developed for the evaluation of qualitative data (e.g., scientific and other texts, interviews, films) in the field of empirical social research. The method was developed to provide researchers with a systematic and intersubjectively verifiable text analysis method that meets scientific standards. The coding guide we designed for this purpose included the categories "intended or recommended ESD goals", "focused educational institution", "recommended means (methods and (teaching/learning) procedures) to promote ESD goals", and "evidence of effectiveness of recommended methods and procedures".

As recommendations for methods and means, we considered suggestions for (a) teaching methods (e.g., experiments, discussions, simulation games, scenario techniques, or lectures), (b) articulation of teaching/learning sequences (e.g., design of teaching phases), (c) teaching/learning procedures (e.g., research-based learning, instructional design), and (d) large-scale forms (e.g., project teaching, problem-based learning). A "special case" was represented by (e) so-called didactical principles (e.g., action orientation, problem orientation, networked, and self-organized learning). These recommendations are very basic and abstract, the formulation of which opens up a wide scope for interpretation. In addition, there are also formulated objectives (personal characteristics to be promoted). The results of the analysis are shown in Table 1. In addition to the recommended methods and the goals to be aimed at with these means, the evidence cited in the articles for their effectiveness is also listed (see Table 1).

**Table 1.** Results of research on means and methods recommendations at the micro level.

| Authorship | (1)<br>(2) | Intended ESD Objectives<br>Educational Institution | Recommended Methods and Procedures for Teaching and Learning | Evidence Provided for the Effectiveness of the Recommended Methods and Procedures |
|---|---|---|---|---|
| (1) Bertschy (2008) [15] | (1)<br><br>(2) | Ability for networked or systems thinking<br>Primary school | Didactical principle:<br><br>- Networked learning with excursions, role plays and exercises for "perspective identification and differentiation", "consideration of consequences" and "perspective merging". | Empirical evidence: results of an efficacy study with a quasi-experimental pretest–post-test design with a treatment and control group. Measurement instrument: game-based guided interview with coding guide. |
| (2) Rieß (2010) [2] | (1)<br><br><br><br><br><br>(2) | Promotion of declarative, conceptual, procedural and strategic knowledge for sustainable development and resource-conserving actions in everyday situations relevant to sustainability.<br>Elementary school | Three steps for the modification of subjective theories (ST) with methods from psychology of volition (will and intention):<br><br>1. Making students aware of the ST that guides their actions (including verbalizing one's actions, changing one's perspective).<br>2. Modification of the ST by adding specialized knowledge (through direct instruction: including presentation of the learning content, guidance of the learning process, practice and consolidation, feedback on goal achievement, stimulation of transfer).<br>3. Transfer of the new knowledge into action-guiding structures (e.g., by making resolutions, stimulation of self-commitment, visualization of action results, use of reminders, and repeatedly reflected action). | Empirical evidence: results of an efficacy study with a quasi-experimental pretest–post-test design with two experimental conditions and a control condition. Quantitative observations and reconstruction of subjective theories. |
| (3) Frisk and Larson (2011) [16] | (1)<br><br><br><br><br><br>(2) | Systems thinking and an understanding of interconnectedness; long-term, foresighted thinking; stakeholder engagement and group collaboration; action-orientation and change-agent skills.<br>School | Real-world case studies with place-based lessons and activities, interdisciplinary approaches to problem-based learning, visioning exercises, forecasting and backcasting activities, community-based service learning, role-playing activities such as mock citizen jury or conflict resolution, group projects and collaborative activities, experiential lessons including project-based learning, and place-based activities, commitment pledges. | Literature reference |

Table 1. *Cont.*

| Authorship | (1)(2) | Intended ESD Objectives / Educational Institution | Recommended Methods and Procedures for Teaching and Learning | Evidence Provided for the Effectiveness of the Recommended Methods and Procedures |
|---|---|---|---|---|
| (4) Künzli-David and Kaufmann-Hayoz (2008) [17] | (1)<br><br><br>(2) | Ability to negotiate and to make decisions with respect to sustainable development.<br><br>School | General didactical principles:<br>- Action and reflection orientation;<br>- Discovery learning;<br>- Accessibility;<br>- Connection of formal and material learning.<br>Specific didactic principles:<br>- Vision orientation, networking learning, participation orientation. | To justify the recommended methods and procedures, reference is made to (a) a modern understanding of education, (b) a constructivist understanding of learning, and (c) the requirements arising from the regulative idea of sustainability.<br>Further evidence is provided in the form of an illustration based on a series of lessons carried out as part of a project and an assessment of feasibility by teachers involved in the project. |
| (5) Piskernik (2008) [18] | (1)<br><br><br><br>(2) | Energy knowledge (in particular, increasing efficiency in the use of energy, saving energy).<br><br>School | Questioning–developing procedure (a method of asking questions that promote learning in the style of a Socratic dialogue) as well as lectures (Powerpoint presentations) by the teacher. | Empirical evidence: results of an efficacy study with a quasi-experimental pretest–post-test design with an experimental and a control group. |
| (6) Schneider (2018) [19] | (1)<br><br><br><br><br>(2) | Promotion of decision-making competences: ability to evaluate and make decisions in situations relevant to sustainability.<br><br>School | Use of an instructional geogame (location-based smartphone game). | Empirical evidence: results of an efficacy study with a quasi-experimental pretest–post-test design with an experimental and a control group condition. |
| (7) Sprenger et al. (2016) [20] | (1)<br><br><br><br>(2) | Promotion of Gestaltungskompetenz (Competence to shape sustainable development) with 12 sub-competences.<br><br>School | - Simulation (experimental) game;<br>- Mystery;<br>- Future Workshop;<br>- Philosophizing with children and young people;<br>- Excursion, visit to extracurricular learning centers. | Interviews with 10 experts (the frequency of naming suitable methods was counted) and review of literature |
| (8) Brundiers and Wiek (2017) [21] | (1)<br><br><br><br><br>(2) | Professional skills in effective and compassionate communication, collaborative teamwork, and impactful stakeholder engagement.<br><br>University, Higher Education | "Constructive alignment and backward design" (a combination of problem-based learning (PBL) and portfolio work, according to Biggs 1999)<br>(among other things, discussion sessions with guest speakers, practical phase (e.g., practicing skills in the classroom, applying skills in a real-world project), text work, role plays, and reflection on the acquisition of skills. | "These concepts ensure that learning objectives are matched with effective learning settings and with appropriate assessments to measure students' progress towards the objectives" (S. 7)<br>Empirical accompanying study: ex post design, 44% of the participants took part in the evaluation. Dependent variable = assessment of own learning gain. |

**Table 1.** *Cont.*

| | Authorship | | (1)<br>(2) | Intended ESD Objectives<br>Educational Institution | Recommended Methods and Procedures for Teaching and Learning | Evidence Provided for the Effectiveness of the Recommended Methods and Procedures |
|---|---|---|---|---|---|---|
| (9) | Fanta et al. (2019) [22] | | (1)<br>(2) | Ability for systems thinking<br>University, Higher Education | Model Problem-Oriented Teaching and Learning (MoPoTL) to promote problem-solving skills relevant for a sustainable development. | Empirical evidence: Results of an efficacy study with a quasi-experimental pretest–post-test design with three experimental conditions and a control condition. Measurement instrument: 4 scales with 23 items in total to capture facets of systems thinking. |
| (10) | Holdsworth and Thomas (2016) [23] | | (1)<br><br><br><br><br><br><br><br><br><br><br>(2) | Competencies and skills in engaging in reflective practice, lifelong learning, critical enquiry and understanding of systems theory (systemic thinking), clarification and judgment of values; critique of ideology; critical reflection and creative thinking; envisaging of sustainable futures; communication skills; creative thinking; personal and professional self-reflection; holistic thinking.<br>University, Higher Education | Experimental and cooperative learning in inter-disciplinary teams, critical (personal and professional self) reflection, sensory and empathic exercise, envisaging of sustainable futures, and experiential learning by reconnecting to emergent situations. | Case Studies. Conclusions were drawn on the basis of semi-structured interviews, surveys, participant observation, and document analysis. Individual statements by participants are cited as evidence. |
| (11) | Juárez-Nájera et al. (2006) [24] | | (1)<br><br><br><br><br><br>(2) | Systems thinking, participatory planning, managing personal and social responsibility, being able to work in multidisciplinary teams and to share own experience with new generations, problem-solving competencies.<br>University, Higher Education | Experiential Learning Cycle: including project-based teaching, working-group-based education, and community-based teaching and education through joint problem definition and joint problem solving. | "According to the participating students, the new course is a success. At the end of the first round of teaching an evaluation showed very enthusiastic responses" (S. 1037) = self-reported data from participating students. |

**Table 1.** *Cont.*

| Authorship | (1)(2) | Intended ESD Objectives / Educational Institution | Recommended Methods and Procedures for Teaching and Learning | Evidence Provided for the Effectiveness of the Recommended Methods and Procedures |
|---|---|---|---|---|
| (12)  Lozano et al. (2017) [25] | (1) | Fostering 12 ESD competences (among others): Systems thinking, anticipatory thinking, ability to work on complex problems in interdisciplinary contexts, critical thinking and analysis, empathy and change of perspective, strategic action, and tolerance for ambiguity and uncertainty. | Pedagogical approaches (among others):<br><br>-    Case studies;<br>-    Interdisciplinary team teaching;<br>-    Lecturing;<br>-    Mind- and Concept-Mapping;<br>-    Project work and problem-based learning (PBL);<br>-    Community service learning;<br>-    Jigsaw method;<br>-    Participatory action research;<br>-    Place-based environmental education;<br>-    Supply chain and life-cycle analysis. | Analysis of texts using methods of hermeneutics and grounded theory. Evidence is ensured by several rounds of discussion among the five authors. |
|  | (2) | Higher education | | |
| (13)  Seatter and Ceulemans (2017) [26] | (1) | Developing the capacity for recognizing and understanding the complexity of sustainability issues, and for thinking critically about assumptions, biases, beliefs, and attitudes while actively participating in their resolutions. Autonomous, creative, critical, and transformative thinking, feeling, and acting. Competence to change perspectives | Transformative learning:<br><br>-    Asking critical questions;<br>-    Constantly searching for new sources and ideas;<br>-    Student-centered rather than teacher-centered;<br>-    Authentic field excursions;<br>-    Case studies;<br>-    Active and constructivist learning;<br>-    Inquiry methodology;<br>-    Community service learning. | Literature reference. |
|  | (2) | University, Higher Education | | |
| (14)  Segalàs et al. (2010) [27] | (1) | Promotion of knowledge, abilities, values and attitudes needed to contribute to SD. Promotion of critical thinking, systemic thinking, and the ability to work within transdisciplinary frameworks. | Learning in groups and with constructivist methods for active learning: e.g., problem-based learning (PBL), case studies. | Pre-experimental research design with multiple groups (no systematic variation of experimental conditions, confounding of multiple variables: Teacher, content, timing, method). Data collection through concept maps created by students in 15 min (number, relevance, linkages, and complexity). |
|  | (2) | University, Higher Education | | |

Table 1. *Cont.*

| Authorship | (1)<br>(2) | Intended ESD Objectives<br>Educational Institution | Recommended Methods and Procedures for Teaching and Learning | Evidence Provided for the Effectiveness of the Recommended Methods and Procedures |
|---|---|---|---|---|
| (15)   Tejedor et al. (2019) [28] | (1)<br><br><br><br><br><br><br>(2) | Development of skills for problem solving such as systemic and anticipatory thinking, critical and creative thinking, capacity for strategy and action, and the collaborative skills of graduates as agents of change for sustainability.<br>University, Higher Education | Active learning strategies:<br><br>-   Service learning;<br>-   Problem-based learning;<br>-   Project-oriented learning;<br>-   Simulation games;<br>-   Case studies. | The effects of the recommended means are assumed because (a) the methods and procedures are compatible with or derived from the constructivist learning paradigm, (b) ten researchers involved in a research project (EDINSOST—education and social innovation for sustainability) agreed on this selection of methods and procedures. |
| (16)   Molitor (2016) [29] | (1)<br><br><br><br><br><br><br><br>(2) | Development of Gestaltungskompetenz (Competence to shape sustainable development) with 12 sub-competencies and appreciation for other people including future generations, for diversity and variety, for other living communities and for raw materials.<br>Adult education | Didactical principles:<br><br>-   Principle of self-organization and self-determination,<br>-   Working on real-life problems, e.g., through projects in the local environment,<br>-   Participation, interdisciplinarity, self-reflection.<br><br>Methods:<br>Future workshop, role-plays, planning games, explorations, experiments, dialogical procedures, conversations, theater projects. | Reference to model projects. |
| (17)   Rieckmann (2016) [30] | (1)<br><br><br><br><br><br><br><br>(2) | Promotion of key competencies for ESD:<br>-   Competence in networked thinking and dealing with complexity;<br>-   Competence to think ahead;<br>-   Competence to think critically, as well as the ability to engage in a critical discourse on values.<br>All educational areas | Participatory and collaborative forms of problem-based learning:<br><br>-   self-organized learning;<br>-   discovery learning;<br>-   inquiry-based learning;<br>-   project-based learning (projects in serious situations);<br>-   promotion of participation and reflection;<br>-   multi-perspectival and interdisciplinary thinking and working. | Literature reference. |

## 4. Results

Seven of the analyzed articles addressed ESD in schools (primary or elementary schools; age 5–11/high schools (= secondary education; age 11–17), eight articles address ESD in higher education (higher or postsecondary education, e.g., at universities; age > 18), one article addressed ESD in adult education, and one article addressed ESD in all of the stated educational sectors. With regard to our first research question, we found that numerous authors expressed the conviction that ESD needs new, modern, alternative, and innovative procedures and methods—some even speak of a new learning culture or pedagogy—to achieve its goals [16,17,20,23,26–28,30].

What are the methods or didactical principles recommended by the authors included in this analysis?

The overwhelming majority of authors suggested only methods and procedures that are compatible with the constructivist learning paradigm for promoting comprehensive ESD goals [31]. In this view, self-regulated and self-directed learning of applicable knowledge and problem-solving skills are assigned a central role [32]. Within this paradigm, knowledge acquisition is seen as a function of active and self-regulated knowledge construction and transformation rather than a result of the acquisition and accumulation of knowledge [33]. Representatives of constructivist approaches recommend methods with no or only weak guidance by the teacher, who has a more moderating role. In the analyzed articles, the corresponding methods included problem-based learning (PBL), portfolio work, role-playing and simulation games, real-world case studies, project-based learning (project work), experiential learning, cooperative learning, community-based service learning, participatory action research, future workshop (envisaging of sustainable futures), inquiry and discovery learning, transformative learning, and authentic field excursions.

The following 11 articles can be counted among this group: Brundiers and Wiek 2017 [21]; Frisk and Larson 2011 [16]; Holdsworth and Thomas 2016 [23]; Juárez-Nájera et al., 2006 [24]; Künzli David and Kaufmann-Hayoz 2008 [17]; Molitor 2016 [29]; Rieckmann 2016 [30]; Seatter and Ceulemans 2017 [26]; Segalàs et al., 2010 [27]; Sprenger et al., 2016 [20]; Tejedor et al., 2019 [28].

Six articles tended to align their means and methods recommendations with the cognitivist view of learning, or they recommended methods that can be categorized partly in one learning paradigm and partly in another: Bertschy 2008 [15]; Fanta et al., 2019 [22]; Lozano et al., 2017 [25]; Piskernik 2008 [18]; Rieß 2010 [2]; Schneider 2018 [19]. The cognitivist perspective on learning emphasizes the construction of knowledge and individual information processing. Declarative and procedural metacognitive knowledge, strategy knowledge, and conditional knowledge from sustainability science are then seen as necessary prerequisites for sustainable action. A unifying commonality of cognitivist approaches is the idea of "meaningful learning", that is, reflected and insightful learning. Teachers can promote such meaningful learning by presenting factually well-structured knowledge to their students and encouraging them to deepen and consolidate this knowledge through application, transfer, and problem solving. The field in which we are interested is mainly about knowledge that is needed to solve problems in sustainability contexts. This teaching strategy is also referred to as direct instruction [34]. The learning activity is largely designed and guided by the teacher, who, among other activities, selects the learning content, plans the course of instruction, and designs and presents the necessary content. In addition, the teacher provides problems to be solved to increase the level of learner activity and autonomy within the framework of exercises and tasks [35]. Suitable procedures and methods recommended by the authors of the included articles include lectures, presentations, class discussions, the model of problem-oriented teaching and learning, questioning–developing procedure, modification of subjective theories, guidance of the learning process, and the stimulation of transfer.

Our second question addresses the evidence the published studies present for the effectiveness of the recommended methods. All articles cite academic literature in support of their recommendations. Some articles also reported on negotiation processes and discus-

sions among authors [25] or among 10 collaborators in a research project [28], as well as results of an expert survey [20]. Articles also reported on model projects [29], case studies with participant observations and document analyses [23], and an implemented series of lessons [17].

A larger group reported the effectiveness of the recommended methods based on (more or less) controlled empirical studies. These studies use study designs classified as (a) a pre-experimental design (one-shot studies; without systematic variation of experimental conditions and without a control group) [27], (b) an ex post facto design (without pretest and also without a control group in the present case) [21], and (c) quasi-experiments, with one or more experimental conditions as well as a control group [2,15,18,19,22].

## 5. Discussion and Preliminary Recommendations

Our first question addressed the recommended means in order to achieve the goals of ESD. A considerable number of recommendations assumed that a successful ESD (realizing its goals) requires new, modern, alternative, and innovative procedures and methods. To promote ESD goals, the corresponding articles predominantly proposed methods that should enable self-regulated and self-directed learning of applicable knowledge and problem-solving skills and a new learning culture or pedagogy oriented towards the constructivist learning paradigm. A smaller group, in contrast, also recommended methods that follow a more cognitivist view of learning and emphasized a higher necessary degree of guidance by the teacher. As a result, teachers in schools and universities in particular are faced with the question of which of these recommendations they can trust the most in terms of their effectiveness and which methods they should prefer to use in their teaching. Our second question addressed the evidence for the effectiveness of the recommended methods in promoting ESD goals. Comparing the evidence provided by the group that favored methods with an emphasis on self-direction and the evidence provided by the group that recommended methods in which a higher degree of guidance of the learning process is attributed to the teachers, we can see an interesting tendency, at least in the articles we examined, which we will report below.

The first group recommending methods with a high degree of self-direction cited the results of studies in the form of self-reports and self-assessments by participants in seminars and model projects (students, teachers) and expert surveys as evidence of the effectiveness of the methods they propose. In addition, the results of two empirical studies were presented as evidence of effectiveness. One of these studies was conducted within an ex post design [21] and the other within a pre-experimental research design [27]. From a research-methodological and scientific-theoretical perspective, all of these types of studies are of an exploratory (investigative) nature [36]. Such studies are informative when exploring new scientific areas and creating theoretical or conceptual preconditions for initial hypothesis formulations. However, they are not suitable for testing an assumed effectiveness of means and methods.

In the second group, which recommends methods with a high degree of guidance by the teacher, evidence for the effectiveness of methods and procedures is provided by references to hypothesis-testing, quasi-experimental studies [2,15,18,19,22]. In contrast to the non-experimental research methods stated above, quasi-experimental studies can be used for testing hypotheses and thus provide evidence for the effectiveness of methods. Experimental studies in which people are randomly assigned to conditions would be even more reliable, allowing conclusions to be drawn about causal relationships. Experimental studies, however, are difficult to realize in educational contexts because they investigate natural groups nested in classes [37]. In addition, the validity of quasi-experiments can also be increased with experimental control (e.g., pre–post or control-group test design, two or more treatment groups, control of possible confounding variables, and documentation of the reliability of the measuring instruments) [38]. In sum, examining the quality of the quasi-experiments in the group of studies oriented towards the cognitive learning

paradigm, we can speak of means recommendations with comparatively good proof. In any case, the evidence for the recommendations is higher than in the first group.

We can thus summarize that in the articles examined, the methods with a higher degree of guidance by the teacher tended to have a more valid claim for effectiveness. For educators (e.g., teachers at schools and universities), this conclusion means that they can justifiably place more weight on these recommendations than the recommendations from the first group. Thus, choosing these learning and teaching methods for their classes would be well justified. Our conclusions, however, do not mean that these methods are more effective than the methods with a higher degree of self-direction or even that the latter are ineffective. To be able to make a comparative assessment of effectiveness, these different methods would need to be systematically tested in the same study. Moreover, different age groups, learner characteristics, and different target dimensions in sustainability competence should be taken into account. A first step in this direction was taken by Asshoff et al. [39], who compared different instructional settings for the use of a learning app in the context of climate change education for student teachers. In conclusion, the literature lacks high-quality effectiveness studies, which should especially provide teachers in schools and universities with the secure knowledge of effective means for promoting ESD goals.

Despite the dearth of ESD research, the results and findings of general empirical educational research (especially psychological research on teaching and learning and teaching research on effective teaching procedures) can be used to derive subject-specific teaching procedures [32]. This proposal is based on the obvious assumption that the acquisition of desired personal characteristics (e.g., complex knowledge, problem-solving skills, and motivational orientations) in ESD does not fundamentally differ from the acquisition in other educational areas. Based on this assumption, we make the following recommendations [40].

*5.1. Methods and Procedures for Promoting Knowledge and Problem-Solving Skills*

The following methods have shown to be effective for building knowledge and promoting problem-solving skills in a variety of ways [32,34]:

- The starting point for learning should be a real-world problem from the sustainability context.
- Acquiring sustainability-relevant knowledge and activating prior knowledge are fundamental.
- New sustainability knowledge or new solutions to problems should be presented.
- Independent phases of learning and problem solving are important but should be supported by feedback and assistance.
- Reflection on one's learning process is significant in promoting increasingly self-directed learning.
- Exercises and also the development of routines foster automation and thus promote effective processing of problem solutions.

More specifically, the following evidence-based recommendations for methods can be made:

For beginner learners, they lack in-depth knowledge and competence (expertise) in a specific subject or a content domain (e.g., global warming, mass tourism, biodiversity crisis), methods with a higher degree of guidance by the teacher (e.g., direct instruction and a problem-based teaching and learning model) might be the first choice for promoting the development of knowledge and problem-solving skills (e.g., systems thinking). With increasing expertise, fostering more extensive self-direction using appropriate methods (e.g., project work, discovery learning) becomes more important, especially due to motivation psychology [41–43]. However, independently of the methods used, educators should be aware of the expertise-reversal effect. Support measures that are beneficial for learners with low levels of prior knowledge (e.g., scaffolding prompts, and worked examples) lose their effectiveness and sometimes even have detrimental effects for learners with higher levels of prior knowledge [44].

*5.2. Methods and Procedures for Promoting Motivation and Attitudes*

To promote personal characteristics with high affective–motivational components (e.g., attitudes toward sustainable development, acceptance and adoption of the maxim of inter- and intragenerational justice, willingness to participate, and empathy), only "relatively effective" means are possible. That is, only under certain circumstances, the means are more likely to promote the desired characteristics [4]. The reason for this uncertainty lies in the complexity of the human personality, which eludes all-too-simple influencing of central personality traits. Inducing cognitive conflicts, stimulating perspective taking, confronting the learner with divergent arguments, and enabling the experience of competence and autonomy have proven to be helpful [41,45]. Potentially suitable methods that can facilitate the promotion of motivation and attitudes include role playing, simulation games, learning from models (observation and imitation learning), value clarification, projects and internships in contexts relevant to sustainability, and the formation of student parliaments in which the learners participate in decisions on matters relevant to sustainability.

*5.3. Methods and Procedures for Promoting Behavioral Readiness*

First, a brief preliminary remark: Theories of action from the field of psychology are based on the basic assumption that various forms of knowledge and motivational factors (including subjective and social norms, attributions of responsibility) can interact and lead first to the formation of behavioral intentions and then to behavior that is relevant to sustainability [46]. In addition to these internal factors, external conditions (e.g., behavioral offers, situational conditions, social norms, and lifestyle of the social environment) also influence sustainability-relevant behavior. Thus, by promoting knowledge and motivational orientations, in turn, the desired behavior can also be promoted. Nevertheless, research has repeatedly demonstrated a considerable gap between knowledge, motivational orientations, and actual sustainability-promoting behavior [47].

Moreover, conflicts can also occur between our knowledge, motives, and behavior, such that the discordance can cause cognitive dissonance [48]. For example, a person may know that air traffic is a major contributor to greenhouse gas emissions and that reducing air traffic would be a sustainable behavior. However, this person has always dreamed of visiting a country abroad, but this action would conflict with the sustainable behavior of not flying. To reduce the dissonance, people usually "reinterpret" their knowledge to justify their behavior. For example, the person wanting to fly abroad might argue that the plane would fly anyway. Thus, no responsibility would be assumed, and the trip would be justified. To promote changes in behavior, teachers at schools and universities should raise awareness (and problematize) their students' action-guiding ideas and assumptions (subjective theories) and foster their self-efficacy so that they believe their actions have an effect. Teachers can achieve this change, for example, by supporting the formation of concrete action resolution, stimulating self-commitment, visualizing action results, and using reminders [2].

Finally, we want to draw attention to a misconception that is also quite common in the ESD literature. It consists of "unjustifiably dividing learning arrangements into those that promote either passive or constructive and active learning" [49] (p. 211). The assumption is that only so-called constructivist methods with a high degree of self-direction by the learners would enable constructive and active learning. However, methods characterized by a high degree of guidance by the teacher (e.g., traditional teaching–learning formats such as lectures, direct instruction, and teacher-centered classroom discussion) have been widely shown to achieve a high level of cognitive activation if designed and implemented thoughtfully [49]. In contrast, methods considered "active" (e.g., group work, learning with computer simulations) might fail to cognitively activate the learners if designed poorly. Thus, there is no "one" method for stimulating learning processes; rather, the effectiveness of a method depends on its design and implementation. Against this background, a high demand currently exists for empirical effectiveness studies that focus on teaching and learning in the field of ESD, and more specifically, evidence-based recommendations of how

ESD goals can be effectively promoted are urgently needed. The formulation of apparently plausible and modern-sounding recommendations for methods is not a sufficient substitute for this need.

## 6. Conclusions

At the very end of this study, the following can now be stated. Within the framework of ESD, numerous goals are proposed in the form of personal characteristics that are to be promoted among learners in schools and universities. For many of these personal characteristics, it has been proven that they actually enable people to contribute to sustainable development. So today we know pretty well what basic knowledge, attitudes, skills, competencies. and values people need in order to actively participate in shaping sustainable development in different contexts and in different places.

In our study, we started at this point and investigated the question of how teachers should teach in order to support the development of these personal characteristics as effectively as possible. For this purpose, we analyzed 17 scientific publications in which suggestions for teaching methods in ESD were made. These suggestions could be assigned to two large groups. The first group recommends methods with a high degree of self-direction, and the second group recommends methods with a high degree of guidance by the teacher. The scientific evidence put forward in support of each recommendation varies widely in quality. All in all, on the basis of this scientific evidence, we are currently unable to provide teachers in schools and universities with clear guidance on how to design effective lessons to promote ESD goals. At this point, however, we can draw on the findings of empirical research on effective methods from other areas of education. We have presented a good range of these methods for which effectiveness has been demonstrated fundamentally and in different settings (cultures, places, content areas). Nevertheless, ESD research should in the future increasingly turn to the question of effective methods in order to be able to provide teachers at schools and universities with increasingly reliable findings on effective methods. This is an important prerequisite for ensuring that the desired ESD goals are actually achieved.

**Author Contributions:** Conceptualization, W.R. and C.M. Methodology: W.R., C.M. and H.-G.K.; Analysis: W.R. and M.M.; Interpretation of data and writing: W.R., M.M., C.M., H.-G.K. and E.-M.W.; Original draft preparation W.R. and M.M. All authors have read and agreed to the published version of the manuscript.

**Funding:** This research was funded by the Baden-Wuerttemberg Ministry of Science, Research, and Culture and the Ministry of Education, Youth and Sports Baden-Wuerttemberg.

**Informed Consent Statement:** Not applicable.

**Conflicts of Interest:** The authors declare no conflict of interest.

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
