# Peer review of "How Can Education for Sustainable Development (ESD) Be Effectively Implemented in Teaching and Learning? An Analysis of Educational Science Recommendations of Methods and Procedures to Promote ESD Goals"

_sustainability, doi:10.3390/su14073708_

Round 1

Reviewer 1 Report

Authors are reviewing 17 paper related to the education for sustainable development. There are several issues that should be addressed

1- English language of the paper is not fluent. Reader would face some difficulties. English language editing and polishing should be done.

2- Introduction is too short. Discuss importance of the topic, research gaps, and your contribution. The motivation to do a literature review should be completely explained in the introduction.

3- In section 3, all databases used for the searches should be mentioned.

4- Research methodology should be explained better. A flowchart also can be a good option.

5- Result section needs more elaboration on findings.

Author Response

Dear Reviewer 1,

We would like to express our sincere gratitude for the attentive reading of our article and the astute and helpful suggestions for editing. We have managed to address all the comments and we hope that the proposed solutions are convincing.

Authors are reviewing 17 paper related to the education for sustainable development. There are several issues that should be addressed

1- English language of the paper is not fluent. Reader would face some difficulties. English language editing and polishing should be done.

Our response:

Thank you for this advice, which we have gladly followed. We have commissioned a native-language proofreader. He polished the whole text in terms of language and brought it up to scratch. Since numerous corrections have been made, we have not highlighted them to maintain the readability of the text.

2- Introduction is too short. Discuss importance of the topic, research gaps, and your contribution. The motivation to do a literature review should be completely explained in the introduction.

Our response:

Yes, here we were far too brief and simply assumed too much in the way of background knowledge on the part of the reader. Now we have worked out the significance of the topic even more clearly, pointed out the research gaps and outlined our intended contribution.

3- In section 3, all databases used for the searches should be mentioned.

Our response:
Thank you very much for this hint. The sentence was formulated in a misleading way. We have mentioned all databases used for the search in the text. We have now made it clearer.

4- Research methodology should be explained better. A flowchart also can be a good option.

Our response:
Yes, that was quite a great suggestion. We constructed a flowchart that should give a good overview of our approach.

----------------------------------

Here, insert Figure 1 The searching process.

------------------------------------

5- Result section needs more elaboration on findings.

Our response:
We have expanded the results section.

Reviewer 2 Report

The article is interesting as perspective "meta" about some issues. However, it should be improved in some parts. First of all it should be declared (possibly in the introduction or in the second paragraph) which is the used approach: it seems psychological, but that is not so clear. Linked to this issue, it is important to clarify the implications of a "meta" reasoning for the starting discipline. Secondly, it is important to clear "which" is the method used to analyze the considered texts (especially in paragraph 3). About this, it would be interesting to know if the authors can use a semiotic approach to the texts. In this regard, it would be interesting to know if the authors can apply a semiotic approach to the texts, in order to give the analysis more strength and, above all, in order to identify some significant nuclei. Finally, bibliographic references should be more international: there are a lot in German language, and it can restrict the readership.

Author Response

Dear Reviewer 2,

We would like to express our sincere gratitude for the attentive reading of our article and the astute and helpful suggestions for editing. We have managed to address all the comments and we hope that the proposed solutions are convincing.

1 - The article is interesting as perspective "meta" about some issues. However, it should be improved in some parts.

Our response:
Thank you very much for the overall very interested and friendly review about which we were very pleased.

2 - First of all it should be declared (possibly in the introduction or in the second paragraph) which is the used approach: it seems psychological, but that is not so clear.

Our response:
Yes here we were not clear enough in our explanations. We have now introduced the working group and our professional background as well as our used approach in the introduction section.

3 - Linked to this issue, it is important to clarify the implications of a "meta" reasoning for the starting discipline.

Our response:
Again, yes we provided too little background information and did not sufficiently address necessary aspects. Therefore, we have now also named the goals and implications of our study for ESD and our subjects in the introduction.

4 - Secondly, it is important to clear "which" is the method used to analyze the considered texts (especially in paragraph 3).

Our response:
We have described in more detail the method chosen in the study to analyze the articles.

5 - About this, it would be interesting to know if the authors can use a semiotic approach to the texts. In this regard, it would be interesting to know if the authors can apply a semiotic approach to the texts, in order to give the analysis more strength and, above all, in order to identify some significant nuclei.

Our response:
Only one member of our research team has heard of semiotics and its methods. Unfortunately, none of us is able to work accordingly. However, we acknowledge that this would be a very interesting and certainly more advanced approach to gain further insights from the analysis of the selected articles. We hope that this work will be tackled by more knowledgeable researchers than us.

6 - Finally, bibliographic references should be more international: there are a lot in German language, and it can restrict the readership.

Our response:
Thank you for this important hint with which you are absolutely right. Where possible we have now replaced German language articles with English language articles. We succeeded in doing so in five cases.
We succeeded in doing so in five cases. Where we were unable to do so, we at least translated the German titles into English so that readers could get a rough idea of what the articles were about.

Reviewer 3 Report

See attached comments.

Author Response

Dear Reviewer 3,

We would like to express our sincere gratitude for the attentive reading of our article and the astute and helpful suggestions for editing. We have managed to address all the comments and we hope that the proposed solutions are convincing.

Lines:

60-61 – This is a small point, but is it “promotion of sustainability competence…” or is it sustainability competence? I ask this because in your next sentence, you use just the latter term (even though I recognize you are citing different sources/authors.) Again, a small point, but shouldn’t it be sustainability competence instead of promotion thereof?

Our response:
You are absolutely right. This is really confusing. We have now deleted the term "promotion“.

163-165 - I wonder whether it is worth the time to explicitly describe what these three categories are (since the authors may be speaking from a German educational context. E.g., what is the distinction between “school” and “higher education” and how were those defined in the various articles that refer to them.) And, actually, when I got to Table 1, I do think it’s worth defining what those school levels mean (e.g., provide, maybe, the general age groups associated with those different types of education levels since you use the terms “Primary School,” Elementary School,” and “School.”) I would say that University and Adult are probably self-explanatory. I do think this is important, as the methods described would look very different depending on the age of the students.

Our response:
This note is also very important. We have followed your suggestion and assigned the age of the students to the different types of schools. In the American context, we usually speak of Elementary School when classifying the place of the first basic education, in countries characterized by the school system in Great Britain of Primary School.

190 - I would start a new paragraph at “Six articles tend to…” I would then not start a new paragraph at “The cognitivist perspective…” The only reason I point to this is that the constructivist and cognitivist perspectives are sort of muddled in the way they are presented. I had to go back and re-read just to sort them out.

Our response:
Yes, that is a good suggestion, which we have implemented.

196-197 - Another reference to sustainability promotion. Obviously, I’m still just sort of wrapping my mind around whether this is an important distinction.

Our response:
Yes, this sentence is anything but easy to understand. We have reworded it:

227-289 - Honestly, I was turned around with all the references to “first” and “second” in various contexts and discussions of conclusions related to those references. You might go back and carefully re-read those sections and perhaps label what those firsts and seconds are referring to. One of the things that caused me to realize my issue here was your conclusion reached in lines 272-273. I’m not sure how you came to that conclusion.

Our response:
Thank you very much for this note. We have now made additions and clarifications so that the reader knows what these enumerations refer to without having to go back in the text.

318 - “loose” should be “lose”

Thanks.

338 - I really like this entire discussion on behavioral readiness.

We are very happy about that!

Round 2

Reviewer 2 Report

Thank you for modifying the article according to the suggested changes.

Author Response

Thank you very much for your helpful suggestions!